# Detection of Image Level Forgery with Various Constraints Using DFDC Full and Sample Datasets

**DOI:** 10.3390/s22239121

**Published:** 2022-11-24

**Authors:** Barsha Lamichhane, Keshav Thapa, Sung-Hyun Yang

**Affiliations:** 1Department of Electronic Engineering, Kwangwoon University, 20 Kwangwoon-ro, Nowon-gu, Seoul 01897, Republic of Korea; 2Department of Rehabilitation Engineering, Daegu Haany University, 1 Haanydaero, Gyeongsan-si 38610, Republic of Korea

**Keywords:** sample DFDC, full DFDC, image-detection, light weighted CNN, VGG-19, Xception, inception-ResNet-v2, non-square aspect ratios, variable resolutions

## Abstract

The emergence of advanced machine learning or deep learning techniques such as autoencoders and generative adversarial networks, can generate images known as deepfakes, which astonishingly resemble the realistic images. These deepfake images are hard to distinguish from the real images and are being used unethically against famous personalities such as politicians, celebrities, and social workers. Hence, we propose a method to detect these deepfake images using a light weighted convolutional neural network (CNN). Our research is conducted with Deep Fake Detection Challenge (DFDC) full and sample datasets, where we compare the performance of our proposed model with various state-of-the-art pretrained models such as VGG-19, Xception and Inception-ResNet-v2. Furthermore, we perform the experiments with various resolutions maintaining 1:1 and 9:16 aspect ratios, which have not been explored for DFDC datasets by any other groups to date. Thus, the proposed model can flexibly accommodate various resolutions and aspect ratios, without being constrained to a specific resolution or aspect ratio for any type of image classification problem. While most of the reported research is limited to sample or preview DFDC datasets only, we have also attempted the testing on full DFDC datasets and presented the results. Contemplating the fact that the detailed results and resource analysis for various scenarios are provided in this research, the proposed deepfake detection method is anticipated to pave new avenues for deepfake detection research, that engages with DFDC datasets.

## 1. Introduction

There are hardly any people who do not use computers, smartphones, and other information channels, which has boosted the speed of information transmission, and ultimately the development of the internet. The world is consuming more and more images, videos, and other data via social media such Facebook, Twitter, video mails, and so on, which shows the profound impact of the internet on people’s daily lives. Undoubtedly, the legitimate and germane videos and images are convenient, friendly to watch, and are innocuous. In contrast, illegitimate and spoof videos and images are exacerbating the lives. These pirated videos or images, recognized as deepfakes [1] are inundating the internet at an alarming rate. Many famous faces such as those of celebrities, social workers, politicians, and civilians, have been weaponized by these fake videos [2] for nefarious purposes such as scams and hoaxes, celebrity pornography, election manipulation, social engineering, phony terrorism and many more. These instances have caused various mental, financial and security distresses among the ingenuous public. Therefore, there is a vital need for developing models that can detect these piracies in a more realistic scenario, with utmost accuracy, to prevent these calumnies.

Relevant examples of deepfakes are demonstrated in Figure 1. There are many ways to manipulate videos and images. The most common and conventional operations of creating deepfakes mentioned in [1] are adding, replicating, or removing, which are easily accomplished with ubiquitously-available, image-editing packages, such as FaceApp [3], that provide an accessible platform for creating fake images or videos without any prior technical knowledge and experience. The modern approaches make use of advanced computer graphics approaches and deep learning approaches, such as Generative Adversarial Networks (GANs) [4], autoencoders [5], and other state-of-the-art technology. Conventional manipulations may be easily detected with the naked eye. However, deepfakes created with the assistance of these prodigiously advanced technologies, can achieve impeccable results with better semantic consistency, making them impossible to be identified with human visual systems. Hence, discovering an automatic way is a necessity in order to scout such deepfakes.

Recently, the growing interest in deepfake detection has been advocated for by several research funds, workshops, conferences, and international projects such as Media Forensics Challenge (MFC2018) and Deepfake Detection Challenge (DFDC) [6]. To come up with a practical way to identify such deepfakes, other public media and re-search communities have also been cooperating.

The DFDC dataset is one of the most challenging new generation datasets used for deepfake detection. There are two different versions of DFDC datasets, the first is the sample or preview version, and the second is the full version. Since the full version is too large, it is separated into sample and preview datasets to reduce the research burden. To date, there are a limited number of studies such as [7,8], which mention the usage of DFDC datasets for deepfake detection. All these studies have only exhibited the usage of sample or preview DFDC datasets but have not investigated the results obtained by predicting full datasets using the model trained with sample or preview datasets. There are only scarce studies such as [9,10,11] which have been successful in addressing full DFDC datasets for training and testing, as the dataset is tremendously large to handle. In December 2020, one of the studies [12] reported that 80% of the manipulated videos in the full DFDC training set were accurately chosen, whereas for the black box scenario only 65% of them were accurately classified by the model. Though several studies have used the DFDC dataset, this research alone is not sufficient to explore all the scenarios, such as: (1) how the model performs when it is trained with sample datasets and predicted for full DFDC datasets; (2) how it performs when the full images are resized to different resolutions, rather than resized by cropping, as the manipulation can be carried out not only in the face, but anywhere throughout the images, it is necessary to analyze all the parts of the image thoroughly; (3) how it performs when the full image frame is resized by maintaining 1:1 aspect ratio and 9:16 aspect ratio, where 9:16 is the full image aspect ratio; (4) what is the memory requirement to store all the frames extracted from DFDC full videos, and what is the maximum and minimum number of frames presented in all the videos, and many other scenarios. It is arduous to directly train or research such a huge dataset without prior knowledge or estimation of the nature of the datasets regarding size, its memory requirement, its performance when trained with different approaches and scenarios, and so on. So, we compare the performance of the DFDC datasets among various state-of-the-art pretrained models such as VGG-19, Inception-ResNet-v2, Xception. We propose three custom developed CNN networks and evaluate their performance in the above-mentioned scenarios to locate deepfakes where our proposed model (Model-A, Model-B and Model-C), outperformed the results when compared with these state-of-the-art methods. In addition to this, the proposed model is flexible for any kind of image resolution and aspect ratio, hence outbreaks the boundary for training and predicting within specific resolution or aspect ratio only for any image classification or deepfake image detection task. The experiments and results published in this study would be extremely advantageous for estimating the resources and choosing the direction for carrying out future research in deepfake detection using massive DFDC datasets.

The paper is organized into Five sections. We have already introduced the overview of the paper addressing some related works in the first section. We then introduce the-states-of-the arts methods and proposed methodology for deepfake image detection. Then the experimental procedure and results obtained from the experiments are described in the next section, Section 3. Finally, we discuss the limitations and the direction of the future enhancements in Section 4, and then conclude with a summary of the whole paper in Section 5.

## 2. Methods

In this section, we will discuss our proposed deepfake detection model, and other pretrained model, that we used in our study with DFDC datasets to predict deepfakes.

### 2.1. State of the Art Pretrained Models

VGG-19 [13], Inception-ResNet-v2 [14] and Xception [15] are pretrained models for detecting deepfake images, which have proved their excellent fake-detection capabilities in studies such as [8,16], hence why we chose to follow these pretrained architectures. In our evaluation framework, we follow the same architecture considered in [13,14,15], respectively, for Xception, VGG-19 and Inception-ResNet-v2. However, while training, we modified the top layer of all three models which are summarized as follows:The last fully connected layer (fc2) with 4096-dimensional feature output of VGG-19 model, was updated to accommodate the dense layer having 1024-dimensional feature output. Furthermore, the last output layer was designed in such a way which resulted in two classes (real or fake), rather than 1000 classes.We did not remove or modify any layer of Inception-ResNet-v2. We just enhanced the model by optimizing the last output layer, for predicting two classes only instead of 1000 classes. Apart from this, all the parameters and layers were kept as in case of Inception-ResNet-v2.Unlike in Inception-ResNet-v2, for the Xception model we added one extra dense layer with an output dimension of 1024 before output layer, and after average pooling layer of the Xception model. We also modified the output layer the same as for VGG-19 and Inception-ResNet-v2.

The output layer of all pretrained models was designed with a single unit dense layer with sigmoid activation function, to fine-tune the model for deepfake detection. The sigmoid activation function was preferred over the SoftMax activation function in the output layer, as the classification is binary. The weight of optimized layers was only trained during the training period of these three pre-trained models (VGG-19, Inception-ResNet-v2, and Xception) while the weight of all other layers was kept frozen.

### 2.2. Proposed CNN Model Architecture

To allow for efficient training and experiments, under minimum resource requirement, we developed and made use of our own light-weighted CNN model, which provided superior results over the pretrained models for the cases where pretrained models were trained for top layers only. In this section, we elaborate on the generic layout of the proposed network and the detailed specific configuration in the experiment section.

We developed CNN models referred to as Model-A, Model-B, and Model-C to identify the manipulated images by stacking several building blocks which are described below:

Input images: The proposed CNN model accepts input as arrays of image pixel and the color channel. We used Keras ImageDataGenerator for converting images to array pixels in RGB color channel mode. Each of the pixeled input images is then passed through a series of convolutional layers.

Convolutional layers: This is the key layer of convolutional neural network whose purpose is to detect, or extract, the set of features (high-level and low-level features) available in the input image. These layers consist of linear operation (convolution) and nonlinear operation (activation function). Convolution involves the multiplication of weights, which we can call learnable filters or kernels, with the input. The height and width of the filters is changed by heat and trial method, such as (3 × 3) or (5 × 5) or (11 × 11) or (7 × 7), which we call filter size. The convolutional layer weights are learned by the network during the training phase, which corresponds to the exact features that we want to find in the real and fake images.

During training in forward pass, the kernel slides across the height and width of the image and computes convolution at that receptive region. This produces a two-dimensional activation map that gives the responses of that filter at every spatial position. The sliding size of the kernel is called stride. There was an entire set of filters in each convolutional layer where each of them would produce a separate two-dimensional activation map. These activation maps are stacked along the depth dimension and the output volume is produced. For each input image of size width *(W_cin_)* × height *(H_cin_)* × number of channel (*D*) and *D_out_* number of filters, the size of output volume can be determined by the following formulae:$$W_{Cout}=\frac{W_{cin}-F_{C}+2P}{S_{C}}+1$$
$$H_{Cout}=\frac{H_{cin}-F_{C}+2P}{S_{C}}+1$$
where, *S_c_* is stride size, *F_c_* is the size of filter used and *P* is the padding size (which is zero in our case). This would provide the output volume of size *W_cout_* × *H_cout_* × *D_out_*.

Normalization layers: The batch normalization layer introduces some additional computations such as mean, standard deviation, scale, and offset factors. Batch normalization allows us to use much higher learning rates and be less careful about initialization, preventing the training from getting stuck in the saturated regimes of nonlinearities. It also acts as a regularizer and accelerates network training 14 times [17].

Nonlinear activation function: The output of linear operation (convolution) is subjected to nonlinear function for further complex operation. Relu activation function was used in the convolutional layer. Other activation functions such as hyperbolic tangent (tanh), sigmoid, softmax, and leaky-Relu can also be used as activation function. Since the classification is binary in our case, we used sigmoid activation function in the outer layer.

Pooling layers: This layer is used for down-sampling the feature maps, to introduce translation invariance to small shifts and distortions. Our network used max-pooling as a type of pooling layer. There would be no learning parameters in the pooling layer, as this only reduces the number of learning parameters and features that are not learnt in this layer, and hence reduces the cost and time for training. For the activation map of size width (*W_pin_*) × height (*H_pin_*) × *D_out_*, the output volume for the pooling layer is determined by the following formulae:$$W_{pout}=\frac{W_{pin}-F_{p}}{S_{p}}+1$$
$$H_{pout}=\frac{W_{pin}-F_{p}}{S_{p}}+1$$
where *S_p_* is stride size, *F_p_* is the size of filter used yielding the output volume of *W_pout_* × *H_pout_* × *D_out_*.

Dropout layers: This layer is used for preventing overfitting in the network.

Fully connected layers: The output of the last convolutional layer is flattened into a one-dimensional array and is connected to one or more fully connected layers. This layer is responsible for determining the relationship between a class and the position of features extracted by convolutional layer in the image. The layer classifies the input image as real or fake and will compute the class scores. We have used only one fully-connected layer which refers to an output layer with dense one to perform binary classification. The activation function applied to this layer was sigmoid. We have also tried using SoftMax, but the sigmoid activation function in this layer gave significant results over SoftMax, in our case.

Using all of these layers, the proposed models, Model-A, Model-B and Model-C, are developed and discussed below:

#### 2.2.1. Model-A Architecture

We built a network model that comprised two convolutional layers and the other five layers consisted of a pooling layer, normalization layer, flatten layer and dropout layer. The first convolutional layer comprised 32 filters with a kernel size of 5 × 5 and a stride size of 1. In the second convolutional layer, we used 64 filters of kernel size 3 × 3 with a stride size of 1. All the convolutional layers were equipped with the Relu [18] activation function for non-linear operations. The output of each convolutional layer was fed to the batch-normalization layer and then to the max-pooling layer, for feature down sampling. The max-pooling layer involved the stride size of 2 × 2. The results obtained from the max-pooling layer were fed to the dropout layer to reduce overfitting. The dropout rates of 0.8 and 0.6 were used for different layers. The network is concatenated with a flattened layer after the last dropout layer, which converts the output into a one-dimensional array for feeding to the output layer. The flattened layer result was connected to the output layer. At the output layer, we used sigmoid activation function with dense one (as the classification was binary i.e., 0 and 1. 0 for fake, and 1 for real). We trained the Model-A structure with DFDC sample datasets and tested it with the sample dataset. The network architecture and model summary of Model-A is shown in Figure 2 and Table 1, respectively.

#### 2.2.2. Model-B

Model-A and Model-B closely resemble each other, except for the change in kernel size, and differences in output shapes. In Model-B, the first convolutional layer used a kernel size of 11 × 11 and the second used 5 × 5. Except for these changes, the other structure was the same as Model-A. This model was trained with the Kaggle sample DFDC dataset and was tested for the sample DFDC dataset and full DFDC datasets. Table 2 and Figure 3 below show the model summary and network architecture of Model-B.

#### 2.2.3. Model-C

Unlike Model-A and Model-B, Model-C resembles a deeper network, as it adds one more convolutional layer. It consists of 14 layers. Among them, three were convolutional layers and the rest of them were used as pooling, batch-normalization, dropout, and output layers. The first convolutional layer used 32 filters, whereas the second subsequent convolutional layer consisted of 64 filters, and finally the third convolutional layer was operated with 128 filters. All three of these convolutional layers used the same filter size of 3 × 3, reducing the size of kernel or filter as compared to Model-A and Model-B. This network also used sigmoid activation function in the output layer and Relu activation function in the convolutional layer. All the layers are connected in the same fashion as in Model-A and Model-B, which is presented in Figure 4. The model was trained and tested with sample datasets. The summary of the model is shown in Table 3.

## 3. Experiments and Results

### 3.1. Experimental Procedure

We used a computer equipped with NVIDIA GeForce GTX 1660 Graphics Processing Units (GPU) for training the deepfake detection model. The codes were written in the Python programming language [19] and involved several Python libraries such as Keras [20], NumPy, OpenCv, sklearn and other Python libraries.

#### 3.1.1. DFDC Sample Dataset Preprocessing

We extracted video frames (which we refer to as image throughout the paper) from Kaggle sample DFDC datasets by using OpenCV library, where we chose image write quality to be 95, and saved those images in separate folders to serve as input while training the model. The entire dataset provided in Kaggle DFDC is too large to train the fake detection system, hence the sample dataset was chosen to train all but one of the proposed models; Model-B was tested using the full dataset.

Some of the fake videos in the DFDC sample datasets did not have a known source video. Such fake videos were not used for collecting the frames for training fake image detectors. We collected 23,094 real images and 17,396 fake images in total from these DFDC sample datasets, where these real and fake images were split into test (20%) and train (80%) using sklearn Python library. All these extracted images were of resolution 1080 × 1920, with an aspect ratio of 9:16 where 1080 is height, and 1920 is width of the image. Each video had a duration of about 10 to 11 s. There were 300 frames per video extracted on average, where minimum frames per video were 297 and maximum frames per video were 302. We could not use test videos provided in the Kaggle DFDC sample dataset for evaluation, as there was no metadata.json file for verification.

#### 3.1.2. DFDC Full Dataset Preprocessing

There is a total of 100,000 fake videos and 19,154 real videos in the DFDC full dataset according to the metadata.json file. All of the videos were grouped in chunks of up to 50 when we downloaded the whole 475 GB zip file. While inspecting the video files, we found that 8 of them were missing, thus only 99,992 were present instead of the 100,000 fake videos from the DFDC full set that was used. We collected image frames from all of these videos using methodology described in 3.1, which took up nearly 2.5 TB of disk space. Since each of the videos had a varying number of frames, ranging between 83 and 601, the total number of extracted real and fake images from the videos were 5,721,288 and 29,731,337, respectively. All of these frames were subjected to our CNN Model-B for testing.

#### 3.1.3. Training Procedure and Parameter Setting for Proposed CNN and Pretrained Models for Deepfake Image Detection

Before starting experiments with the proposed model, we first experimented with different state-of-the art pretrained models such as the VGG-19, Inception-ResNet-v2 and Xception models, and analyzed their performance. For the VGG-19 model, the original image of resolution (1080 × 1920) was resized to 224 × 224, which is the maximum input size limit was fed as the input to train the network. However, the image was resized to 299 × 299 to train with Inception-ResNet-v2 and Xception, where this input size is the maximum possible input size for both the models. Training with these models provided convincing results, however, they were constrained by the input size limits. Hence, we further developed our own model and carried out experiments where we were not bound to any type of fixed image resolution, or fixed aspect ratio constraint.

As mentioned by authors in [21], the full resolution images which do not undergo resizing, are free from any loss of essential features, hence the model can make decisions based on information gathered from all over the images. Thus, we tried to feed a full resolution image, i.e., 1080 × 1920 × 3 pixel—where 1080 is the height, 1920 is the width and 3 denotes RGB (red, green, blue)—channel to our proposed simple light weighted custom CNN network of Model-A, Model-B and Model-C. These three models were also trained with resized image sizes of: 224 × 224, 299 × 299, 135 × 240, 270 × 480 and 540 × 960. These training images were resized from full images directly without cropping. We experimented with our proposed model with the original aspect ratio of 9:16 and the 1:1 aspect ratio.

Initially, the experiments were carried out using Adam optimizer and SoftMax activation function in the output layer, for all these pretrained models and our proposed models, but the networks did not converge. So further experiments were attempted, by replacing SoftMax with sigmoid activation function, however, the network did not converge at all again. We also tried keeping activation function as SoftMax and changed only the optimizer to RMSProp, but again the network failed to converge. Finally, the experiment was performed by replacing SoftMax with sigmoid activation function, and Adam optimizer with RMSProp optimizer together, which provided acceptable outcomes. Though the network started converging after changing the optimizer and activation function in the output layer, the convergence rate of the VGG-19 network was still prolonged. As a result, the network saturated too early resulting in unsatisfactory output. The result of Inception-ResNet-v2 was somehow better than VGG-19, but not significantly better and the convergence rate was still slow. Therefore, we further carried out the experiment with the Xception pretrained model, which achieved better performance in comparison to Inception-ResNet-v2 and VGG-19, but was still insufficient as validation data performance was still poor compared to training data performance. As a result, we approached a simple, custom-developed, convolutional neural network model rather than training with other pretrained models.

All these models were trained up to 60 epochs and the batch size varied according to the input size and CNN model used. The model with best f1-score was chosen to evaluate the performance. The batch size was set to 32 for all the pretrained and proposed models, when the input image sizes were 224 × 224, 299 × 299, 135 × 240, 270 × 480. For other input sizes (540 × 960, 1080 × 1920), Model-A and Model-B used the batch size of 8 whereas Model-C used the batch size of 4. All of the models were saved after every epoch and the performance of the model was analyzed. The best-performing model (Model-B), was chosen for predicting the full DFDC datasets. The performance of all above mentioned models has been shown in the following section.

### 3.2. Performance Evaluations Metrics

The performance of all above-mentioned models has been experimented and evaluated in terms of evaluation metrics such as recall, precision, f1-score, overall accuracy, and AUC (Area Under Curve) where positive class is defined to be ‘REAL’(1), and negative class is assigned to be ‘FAKE’(0). Each one of these metrices is described in the following points below:

True Positive (TP): It indicates the numbers of positive examples classified accurately. In our case the number of “real” images is classified accurately.

True Negative (TN): It indicates the number of negative examples classified accurately. In our case the number of “fake” images is classified accurately.

False Positive (FP) (Type 1 Error): FP is the number of actual negatives classified as positive.

False Negative (FN) (Type 2 Error): FN is the number of actual positives classified as negative.

Accuracy: Accuracy of the model is calculated using the following formula:$$\mathrm{accuracy}=\frac{\mathrm{TN}+\mathrm{TP}}{\mathrm{TN}+\mathrm{FP}+\mathrm{FN}+\mathrm{TP}}$$

Recall: Recall measures how much we predicted correctly, out of all the positive classes. It should be as high as possible. It is calculated as:$$\mathrm{recall}=\frac{\mathrm{TP}}{\mathrm{TP}+\mathrm{FN}}$$

Precision: Precision gives the measure of how many classes are positive, out of all of the positive classes we predicted correctly. It should be as high as possible. It is calculated as:$$\mathrm{precision}=\frac{\mathrm{TP}}{\mathrm{TP}+\mathrm{FP}}$$

ROC: It is a plot of false positive rate (*x*-axis) versus true positive rate (*y*-axis). It is useful to know at what rates the model is recognizing fake images as real, and real images as fake.

F1-score: It is the combination of the precision and recall of a classifier into a single metric, where it combines them by taking harmonic mean, and is calculated as:$$f1-\mathrm{score}=\frac{2\;\times \;\left(\mathrm{precision}\;\times \;\mathrm{recall}\right)}{\mathrm{precision}+\mathrm{recall}}$$

### 3.3. Experimental Results and Discusssion

In this section we present all the results obtained throughout the experiments. The validation and training scores of VGG-19, Inception-ResNet-v2, Xception, and Model-B obtained from the sample dataset are presented in Table 4, whereas those of Model-A and Model-C are shown in Table A1 in Appendix A. The test results obtained by predicting Kaggle full DFDC datasets with Model-B, that is trained with the sample dataset, are presented in Table 5.

Similarly, the training and validation accuracy vs. epoch graph of the VGG-19 model, Inception-ResNet-v2 model, Xception model, Model-A, Model-B and Model-C together, is shown in Figure 5. It is worth mentioning that in Figure 5, the input image size of 1080 × 1920 has been used. The training accuracy vs. epoch graph of the VGG-19 model, Inception-ResNet-v2 model, Xception model, Model-A, Model-B and Model-C, shown in Figure 5, clearly shows the converging rate of the custom developed model and Xception model is faster than VGG-19 and Inception-ResNet-v2 pretrained model, even after replacing SoftMax with RMSProp activation function. Though the Xception network has shown better accuracy while training, it could not overtake Model-B in terms of validation accuracy in almost all the epochs. The validation accuracy and f1-score of the VGG-19 model was limited to 71% and 0.75, while that of best-performing model, Model-B, has rocketed up-to 97% and 0.97. The validation accuracy of the best-performing pretrained model (Xception model), is 6% less than Model-B, and its f1-score is limited to 0.91.

We also compared the training and validation history among various input sizes and various aspect ratios for Model-A, Model-B and Model-C, which is revealed in Figure 6 for Model-B, Figure A1 and Figure A2 for Model-A and Model-C, respectively, and in Appendix A. The comparative performance of the models among various resolution images (shown in Figure 6), has shown the enhancement of model performance when trained with full resolution images, rather than doing so by resizing to various sizes. There was not much difference in the results obtained with the image sizes 224 × 224, 299 × 299, 135 × 240 and 270 × 480. In contrast, the performance scores of the custom model were increased by five to ten percent on average, when the image resolution was maintained between 540 × 960 and original resolution, 1080 × 1920. It is observed that the model trained by maintaining the image aspect ratio (1:1), and original aspect ratio (9:16), has not shown a significant difference in the performance score, but the resolution of the image affected the performance of the model. However, it is not necessary to convert all the images to resolution 1080 × 1920, if the model must be trained or tested using other datasets.

The runtime taken by each model, Model-A, Model-B and Model-C, were noted for all the resolutions in minutes (min). Model-A took 55.93 min per epoch when it was trained with original full-resolution image input. When trained with resolutions such as 299 × 299, 224 × 224, 135 × 240, 270 × 480 and 540 × 960, it took 10.43 min, 10.91 min, 11.2 min, 13.08 min and 15 min per epoch, respectively, on average. Similarly, Model-B took 10.5 min, 11.13 min, 11.53 min, 13.43 min, 24 min and 98 min per epoch on average when the input resolution was 299 × 299, 224 × 224, 135 × 240, 270 × 480, 540 × 960 and 1080 × 1920, respectively. For the case of Model-C, it took 10.55 min, 10.41 min, 11.13 min, 13 min, 19 min and 70 min per epoch on average when the input resolution was 299 × 299, 224 × 224, 135 × 240, 270 × 480, 540 × 960 and 1080 × 1920, respectively. In this sense the run time for Model-A was shorter compared to Model-B and Model-C.

The destitute test results (accuracy = 37.56%, f1-score = 0.26) in Table 5, concluded that the model trained and tested with the sample dataset is not suitable for predicting the full Kaggle DFDC datasets. Hence it would not be a better idea to generalize the model that is trained with the sample dataset. We need to train the CNN model with all varieties of full datasets if we want to get better prediction results for DFDC dataset.

## 4. Limitations and Future Work

In this work, we tested the Kaggle Full DFDC datasets with only one of our proposed models. We have not tested it with all deepfake detector models that we trained with sample datasets. Furthermore, we trained the model with a sample dataset only and predicted results for a full dataset. In future, we can test the Kaggle Full DFDC dataset with Model-A and Model-C that we used in the experiment and analyze the result. However, if resources and time are limited then instead of testing Model-A and Model-C again with full DFDC dataset, we suggest training the Model-B with full DFDC dataset and then implementing it for video detection too, as Model-B has provided the best score among these three models. However, we do not mean that it is not important to test Model-A and Model-C with the full DFDC dataset too. It is advised to train the model with full resolution, rather than to train by cropping or resizing. We also found some videos mentioned in metadata.json were missing, which we have mentioned in Section 3.1.2. We suggest correcting those misplaced video names in the metadata.json file for ease of further research in DFDC datasets. In the case of full datasets, most of the videos have 300 frames per video, but there are also some videos which contain frames of maximum 601 and minimum 83.

This study has performed various experiments and published the results for different settings. However, this paper has not addressed the core reason behind why Model-B has performed better over Model-A and Model-B. We can see the Xception network has performed better than VGG-19 and Inception-ResNet-v2, and at the same time Model-C has not shown satisfactory results over the Xception network. Though, the validation scores of the proposed model are better than pretrained models for so many epochs, the validation score over the epochs for Model-A, Model-B and Model-C fluctuate and are stable for some epochs only. We are not able to explain the main reason behind these issues. Hence, future research can be directed towards studying the cause of all these existing scenarios and optimizing the models accordingly.

Though the proposed model is focused for deepfake detection purposes only in this study, our proposed model can be implemented for many other image classification problems where we can feed any input image, having any aspect ratio and image resolutions. For example, we can test our model in bacteria image analysis, such as lytic bacteriophage detection in sewage water images [22], in future. The paper in [23] has used a multilayer network for cervical cancer diagnosis. We can also use this approach to detect deepfakes in future and compare the results with our proposed model. Additionally, our proposed model can also be used for the cervical cancer diagnosis and to compare the results obtained in the paper [23].

## 5. Conclusions

In this study, we developed and compared different deepfake image detection models with DFDC datasets. Firstly, we compared our proposed model (Model-A, Model-B and Model-C) with various states of the art pretrained models such as VGG-19, Inception-ResNet-v2 and Xception, where the proposed model (Model-B) outperformed the scores of pretrained models. Unlike the other existing research articles, we also trained and tested the model using various resolutions such as 1080 × 1920, 224 × 224, 229 × 299, 135 × 240, 270 × 480, 540 × 960 for DFDC datasets where we included the image sizes by maintaining both 1:1 and 9:16 (original) aspect ratios, hence developing the flexible model for variable resolutions and aspect ratios. The images were resized without cropping to spot the manipulations throughout the images, rather than limiting to spot forgery within the facial region only. Furthermore, we tested the Kaggle Full DFDC datasets using the model that is trained and validated in the sample dataset, and presented the result, which has never been discussed in any previous research works. Additionally, we also provided an overview of the total number of frames contained in full DFDC videos, missing videos, resources taken by the datasets, and many more crucial results within such a massive dataset, by implementing our best-performing proposed model, Model-B. This would be a great benefit for further research that is focused on DFDC datasets for fake image detection, or fake video detection. Thus, this study fulfills its objective of developing and comparing deepfake image detection in various scenarios, and provides the detailed test results performed using full DFDC datasets.

## Figures and Tables

**Figure 1 sensors-22-09121-f001:**
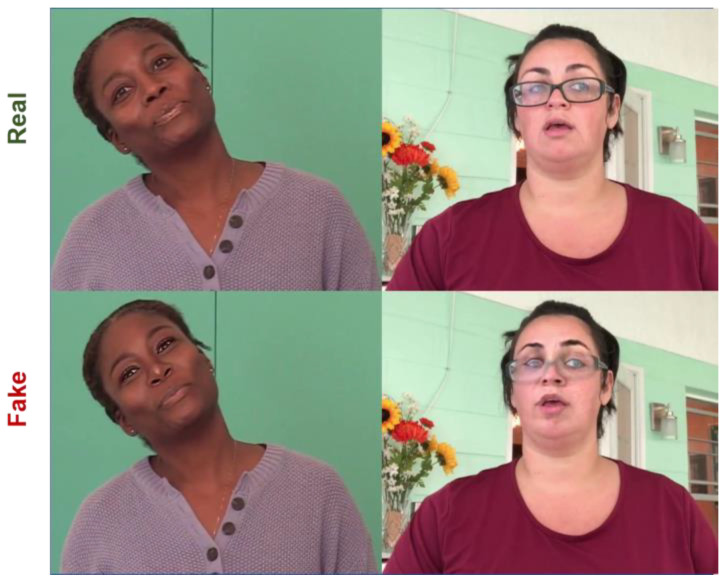
Photo-realistic deepfake images that exist in real life.

**Figure 2 sensors-22-09121-f002:**
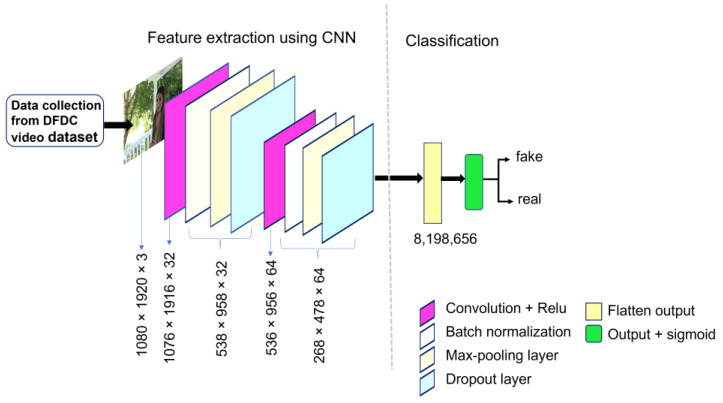
Proposed CNN architecture of Model-A for detecting deepfake images.

**Figure 3 sensors-22-09121-f003:**
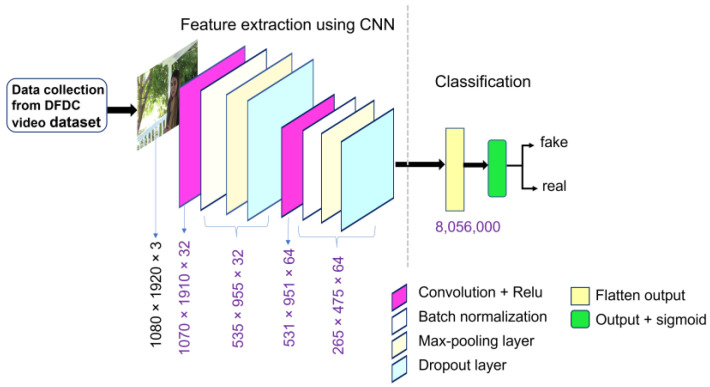
Proposed CNN architecture of Model-B for detecting deepfake images.

**Figure 4 sensors-22-09121-f004:**
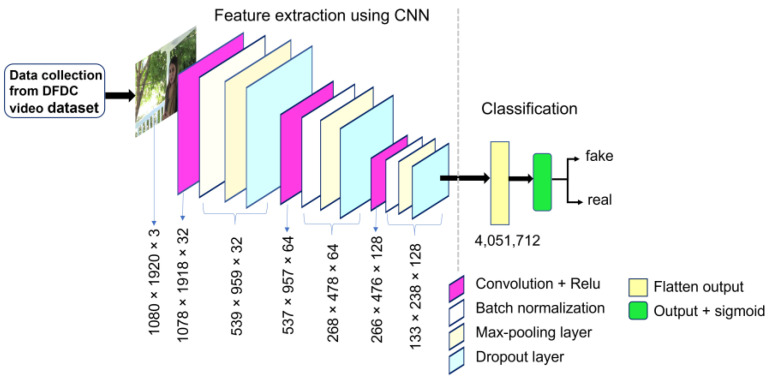
Proposed CNN architecture of Model-C for detecting deepfake images.

**Figure 5 sensors-22-09121-f005:**
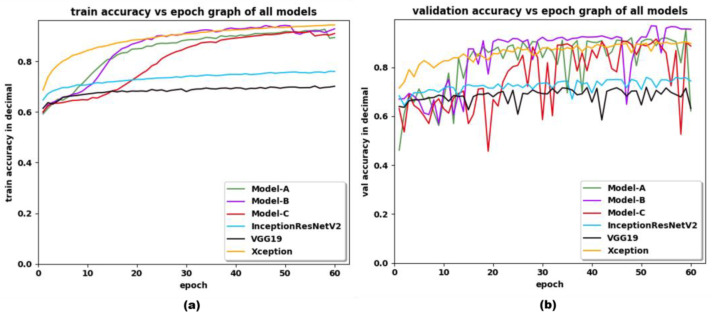
(**a**) Train and (**b**) validation accuracy of Inception-ResNet-v2, Xception, VGG-19, Model-A, Model-B, and Model-C obtained with sample DFDC dataset when trained up to 60 epochs. The input resolution for Model-A, Model-B, and Model-C is 1080 × 1920.

**Figure 6 sensors-22-09121-f006:**
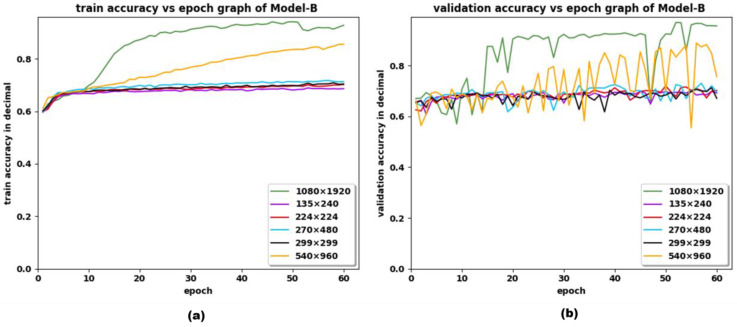
Comparison of (**a**) train and (**b**) validation accuracy versus epoch obtained with various input resolutions 1080 × 1920, 135 × 240, 270 × 480, 540 × 960, 224 × 224, 299 × 299 for Model-B.

**Table 1 sensors-22-09121-t001:** Summary of self-developed CNN network: Model-A for DFDC image detection.

		Model-A			
Layer	Type	Output Shape	Kernel Size	Strides	Dropout Rate
1	Convolution + ReLU	1076 × 1916 × 32	5 × 5	1 × 1	
2	Batch -Normalization	1076 × 1916 × 32			
3	Max-Pooling	538 × 958 × 32	2 × 2	2 × 2	
4	Dropout	538 × 958 × 32			0.8
5	Convolution + ReLU	536 × 956 × 64	3 × 3	1 × 1	
6	Batch-Normalization	536 × 956 × 64			
7	Max-Pooling	268 × 478 × 64	2 × 2	2 × 2	
8	Dropout	268 × 478 × 64			0.6
9	Flatten	8,198,656			
10	Dense + Sigmoid	1			

**Table 2 sensors-22-09121-t002:** Summary of self- developed CNN network: Model-B for DFDC image detection.

		Model-B			
Layer	Type	Output Shape	Kernel Size	Strides	Dropout Rate
1	Convolution + ReLU	1070 × 1910 × 32	11 × 11	1 × 1	
2	Batch -Normalization	1070 × 1910 × 32			
3	Max-Pooling	535 × 955 × 32	2 × 2	2 × 2	
4	Dropout	535 × 955 × 32			0.8
5	Convolution + ReLU	531 × 951 × 64	5 × 5	1 × 1	
6	Batch-Normalization	531 × 951 × 64			
7	Max-Pooling	265 × 475 × 64	2 × 2	2 × 2	
8	Dropout	265 × 475 × 64			0.6
9	Flatten	8,056,000			
10	Dense + Sigmoid	1			

**Table 3 sensors-22-09121-t003:** Summary of self- developed CNN network: Model-C for DFDC image detection.

Model-C					
Layer	Type	Output Shape	Kernel Size	Strides	Dropout Rate
1	Convolution + ReLU	1078 × 1918 × 32	3 × 3	1 × 1	
2	Batch-Normalization	1078 × 1918 × 32			
3	Max-Pooling	539 × 959 × 32	2 × 2	2 × 2	
4	Dropout	539 × 959 × 32			0.8
5	Convolution + ReLU	537 × 957 × 64	3 × 3	1 × 1	
6	Batch-Normalization	537 × 957 × 64			
7	Max-Pooling	268 × 478 × 64	2 × 2	2 × 2	
8	Dropout	268 × 478 × 64			0.6
9	Convolution + ReLU	266 × 476 × 128	3 × 3		
10	Batch-Normalization	266 × 476 × 128			
11	Max-Pooling	133 × 238 × 128	2 × 2		
12	Dropout	133 × 238 × 128			0.25
13	Flatten	4,051,712			
14	Dense + Sigmoid	1			

**Table 4 sensors-22-09121-t004:** Accuracy, recall, precision, AUC obtained with train and validation data (DFDC sample).

		Training Results					Validation Results				
Model	Input Image Resolution	F1 Score	Accuracy (%)	AUC	Recall	Precision	F1 Score	Accuracy (%)	AUC	Recall	Precision
VGG-19 [13]	224 × 224	0.716	68.17	0.751	0.705	0.727	0.751	67.95	0.773	0.849	0.673
Inception- ResNet-v2 [14]	299 × 299	0.789	75.97	0.850	0.790	0.788	0.798	74.41	0.854	0.890	0.724
Xception [15]	299 × 299	0.948	94.15	0.988	0.946	0.950	0.916	90.12	0.967	0.953	0.882
Model-B	1080 × 1920	0.922	91.14	0.936	0.921	0.923	0.973	96.96	0.984	0.989	0.958
	540 × 960	0.856	83.57	0.911	0.856	0.855	0.904	88.83	0.941	0.924	0.884
	270 × 480	0.750	71.75	0.803	0.745	0.756	0.790	73.02	0.823	0.891	0.709
	135 × 240	0.724	68.54	0.771	0.725	0.723	0.761	68.71	0.791	0.875	0.673
	224 × 224	0.727	69.05	0.777	0.723	0.731	0.767	71.24	0.794	0.832	0.711
	299 × 299	0.738	70.42	0.791	0.732	0.744	0.771	69.86	0.804	0.891	0.679

**Table 5 sensors-22-09121-t005:** Test results obtained with DFDC full dataset when predicted with Model-B that is trained with sample DFDC dataset.

DFDC Data Type	Input Image Resolution	F1 Score	Accuracy	Recall	Precision
Full DFDC	1080 × 1920	0.268	0.3756	0.708	0.165

## Data Availability

Not applicable.

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
