# Peer review of "Detection of Image Level Forgery with Various Constraints Using DFDC Full and Sample Datasets"

_sensors, 2022, doi:10.3390/s22239121_

Round 1
Reviewer 1 Report
Detailed Comments
This paper proposes a method to detect deepfake images using light weighted convolutional neural network (CNN) and compare it with various sates-of-the-art pretrained models. To improve the credibility, this paper attempt the testing on full DFDC datasets and presented the results. The experimental results are good, which verify the effectiveness of the model.
The pioneering work of this paper is to conduct experiments under different aspect ratios and resolutions, so that the model can cope with more complex situations. This approach is very interesting.
However, there are some unclear points as follows:
1. The example in Figure 1 is not clear enough
2. Incorrect language exists, such as "excluding introduction section".
3. There is a big problem with the ranking of article structure: there is no content of chapter 2.
4. We have noticed that the test results of the proposed model on the training set and validation set are quite different and fluctuate sharply. Can you explain the reason for this difference?
5. The idea of testing pictures with different resolutions is interesting, but it lacks a rational description.
6. The Inception-ResNet-v2 changes in Section 1.3.2.1 are not clear enough.
7. The paper has the problem of repeated narration. 3.1, 4.1.1 and 4.1.2 discuss the same data set preprocessing process.
8. The network structure of the Proposed three models should be discussed in Section 3.3.2.1 (Proposed CNN model architecture), but the description of the network structure is not detailed enough, and there are too many experimental Settings like image resolution and hyperparameter setting, which is the same as the experimental description in Section 4.1.3.
9. For the comparison of experimental effects of ModelA, B and C, due analysis is lacking in the experimental part. Why Model B based on Inception-ResNet-v2 works best? The effect of Xception is better than that of VGG-19 and Inception-ResNet-v2 (Table2), but why is the effect of Model C based on Xception not satisfactory?
10. There are contradictions in the fifth part. If the author thinks it is unnecessary to test Model A and Model C in the full DFDC Dataset, then it is not included in A Limitation.
Author Response
Foremost, we express our sincere gratitude to the reviewer for his/her insightful and constructive comments on the manuscript. We have thoroughly addressed each issue raised by the reviewer and revised the manuscript accordingly. The underlined texts indicate our responses that are reflected in the manuscript in response to the reviewer’s comments.

Reviewer 2 Report
The main contribution and proposed approach have some novelty in contribution. Revision in terms of technical details is needed before publication. So, some comments are suggested to describe technical details.
1. It is suggested to discuss about the runtime of your proposed approach briefly. (Compare with existing methods is not needed)
2. Add related reference to the compared methods such as VGG and ResNet in the Table 2. Did you re-implement all of the compared methods in the Table 2?
3. Discuss about the novelty of the proposed CNN with more details. Which layers play important role to detect forgery? Brief justification is needed here.
4. The section 3.2.1 is repeated twice in the text.
5. Your proposed approach can be used in many other medical applications such cancer diagnosis or bacteria image analysis. For example, I find a paper entitled “Cervical cancer diagnosis based on modified uniform local ternary patterns and feed forward multilayer network optimized by genetic algorithm”, which has relation. As another example, I find a paper entitled “Isolation and characterization of lytic bacteriophages infecting Pseudomonas aeruginosa from sewage water”. Cite these papers and discuss about it as one of the advantages of your proposed approach.
6. The row and column size of input images in most of benchmark CNN architectures is same. But, these are not same in your proposed CNN as mentioned in the Table 1. What are your reasons? Is it necessary to resize all of the input images to the size of 1080 * 1920?
Author Response

(The authors gave the same response as above.)

Round 2
Reviewer 1 Report
no further comment
Reviewer 2 Report
Most of comments are considered in the revised version. Some descriptions are added which describe the methodology in a clearer way than original submission. The proposed approach has enough novelty to be accepted.